# Associations between Body Mass Index, Waist Circumference, and Myocardial Infarction in Older Adults Aged over 75 Years: A Population-Based Cohort Study

**DOI:** 10.3390/medicina58121768

**Published:** 2022-11-30

**Authors:** So Yoon Han, Nan Hee Kim, Do Hoon Kim, Yang Hyun Kim, Yong Kyu Park, Seon Mee Kim

**Affiliations:** 1Department of Family Medicine, Korea Cancer Center Hospital, Seoul 01812, Republic of Korea; 2Department of Endocrinology and Metabolism, Internal Medicine, Korea University College of Medicine, Seoul 02841, Republic of Korea; 3Department of Family Medicine, Korea University College of Medicine, Seoul 02841, Republic of Korea; 4Department of Biostatistics, Catholic University College of Medicine, Seoul 06591, Republic of Korea

**Keywords:** body mass index, waist circumference, myocardial infarction, coronary artery disease, older adults

## Abstract

*Background and Objectives*: Body mass index (BMI) is widely used as a standard screening method for obesity and an indicator of related diseases. However, its inability to distinguish between lean body mass and body fat limits its utility. This limitation may be more prominent in older populations, wherein age-related sarcopenia and increased visceral fat due to the redistribution of adipose tissue may preclude a precise estimation of obesity. Many studies suggest that waist circumference (WC) is more strongly related to obesity-related diseases. There are also different opinions on whether the obesity paradox is real or a result of confusing interpretations. This study seeks to determine the association between myocardial infarction (MI), BMI, and WC in older adults and to determine if BMI and WC can reliably predict the risk of cardiovascular disease. *Materials and Methods*: We conducted a cohort study of older Korean adults aged over 75 years registered in the National Health Insurance System Senior database. *Results*: The results from the analysis using model 5, which was adjusted for each study variable, showed that the lower the BMI, the higher the hazard ratio (HR) of MI and vice versa. On the other hand, groups with lower than normal WC showed lower HR; even if it was higher, the difference was not statistically significant. Those with abdominal obesity tended to have an increased HR of MI. *Conclusions*: This study found that HR for MI has a negative relationship with BMI, whereas it has a positive relationship with WC. Furthermore, WC is a more appropriate indicator for predicting the risk of MI in the older population.

## 1. Introduction

Obesity continues to increase in global prevalence [1]. Evidence of its association with diverse chronic diseases, such as cardiovascular disease (CVD), type 2 diabetes mellitus (DM), and several cancers, is well-established [2,3,4]. In 2019, the prevalence of obesity in Korean adults over 19 years was 35% based on a body mass index (BMI) of 25 kg/m^2^, and in Korean elders over 70 years, it was 34.3% (men 30.4%, women 37.0%) [5]. Many prospective studies reported that CVD risk increased with obesity [6,7,8,9,10,11,12,13,14,15]. The risk was established in BMI ≥ 30 kg/m^2^, which is considered to indicate obesity according to Western standards. However, some studies reported a similar increase in risk in the overweight (overweight: 25 kg/m^2^ ≤ BMI < 30 kg/m^2^) population [6,7,8,9,10]. In contrast, some previous studies have reported that the risk did not clearly increase or even decreased with increased weight [11,12,13,14,15]; this is termed the obesity paradox. Further, some studies support the obesity paradox [16] and some refute it, citing confounders [17,18]. BMI has been cited as a confounder and is thought to be inadequate for obesity screening [19].

Waist circumference (WC) or WC-to-hip ratio (WHR) has been believed to compensate for the limitations of BMI as an obesity indicator [20,21], especially in the older population with age-related sarcopenia, redistribution of adipose tissue, and increase in visceral fat [22].

This study aimed to investigate the association of obesity with various diseases in older adults aged over 75 years. The traditional cut-off to define older adults, 65 years, has not been used due to a lack of medical basis [23]. Instead, age-related sarcopenia has been shown to be more prevalent in older adults aged over 75 years [24,25]. In Japan, it has been suggested to increase the cut-off to 75 years because of evidence of aging delay by around 5–10 years based on walking speed and grasping power in the past 10–20 years [26].

Many countries today face increasing obesity prevalence as well as a concurrent and steep increase in the population of older adults. Despite many studies describing physical changes during aging, indicators for adults over 20 years were conventionally adapted to the older population regardless of the characteristics in defining obesity and developing guidelines for various diseases caused by obesity.

Therefore, this study aimed to determine if BMI and WC are appropriate indicators for predicting CVD risk in older Korean adults aged over 75 years.

## 2. Materials and Methods

### 2.1. Data Collection

This is a prospective cohort study using the Korean government-operated National Health Insurance System (NHIS) database. NHIS covers 97% of the entire population and recommends a health examination for every adult over 20 years every 2 years. The health examination is conducted in one day and takes 30 min–4 h. Its databases enable cohort studies through data sharing. The NHIS-Senior database includes data selected by a 10% simple random sampling method from participants aged ≥60 years. This study obtained the clinical data of 758,471 older adults from the NHIS-Senior database. Of the 1,018,597 older adults aged over 75 years who have had more than one general health examination provided by the NHIS between 2009 and 2012, 819,342 patients were identified after removing duplicates. Patients with missing information (1675) or who were previously diagnosed with myocardial infarction (MI; ICD I21–I22; 44,304 patients) were excluded. A total of 758,471 older adults (mean age 79.24 years) were finally enrolled in this study. The average follow-up period was 6.94 years until December 2018.

### 2.2. Ethical Considerations

Ethical exemption for this study was granted by the committee of the Institutional Review Board of Korea University Guro Hospital.

### 2.3. Diagnostic Criteria

MI diagnosis was based on ICD-10 code I21–I22 documented during admission for patients who were newly diagnosed during the follow-up period.

DM diagnosis was based on codes E11, E12, E13, and E14 for patients who were receiving hypoglycemic agents or had fasting blood sugar levels ≥ 126 mg/dL.

Hypertension (HTN) diagnosis was based on codes I10, I11, I12, I13, and I15 for patients who were receiving antihypertensive agents or had a systolic blood pressure ≥ 140 mmHg or a diastolic blood pressure ≥ 90 mmHg.

Hyperlipidemia diagnosis was based on code E78 for patients who were taking hyperlipidemia medications or had total cholesterol levels ≥ 240 mg/dL.

Chronic obstructive pulmonary disease (COPD) diagnosis was based on codes J41, J42, J43, and J44.

End-stage renal disease diagnosis was based on special case calculations V001, V003, and V005.

Cancer diagnosis was based on special case calculation V193.

### 2.4. Subject Classification

According to the classification criteria of the Korean Society for the Study of Obesity, BMI was classified as level 1: BMI < 18.5 kg/m^2^, level 2: 18.5 kg/m^2^ ≤ BMI < 23 kg/m^2^, level 3: 23 kg/m^2^ ≤ BMI < 25 kg/m^2^, level 4: 25 kg/m^2^ ≤ BMI < 30 kg/m^2^, and level 5: BMI ≥ 30 kg/m^2^ and the normal level, BMI level 2 was selected as a reference level. The Korean Society for the Study of Obesity defines WC in the normal range for males as WC < 90 cm and females as WC < 85 cm. Unlike BMI, WC does not have a commonly applied range for being underweight, normal weight, or obese. Therefore, we classified WC into 5 levels based on the upper limit of the normal range (males 90 cm and females 85 cm) in 10 cm increments and selected level 3, which is the normal upper limit, as the reference level. WC was classified as level 1: male, WC < 70 cm/female, WC < 65 cm; level 2: male, 70 cm ≤ WC < 80 cm/female, 65 cm ≤ WC < 75 cm; level 3: male, 80 cm ≤ WC < 90 cm/female, 75 cm ≤ WC < 85 cm; level 4: male, 90 cm ≤ WC < 100 cm/female, 85 cm ≤ WC < 95 cm, and level 5: male, WC ≥ 100 cm/female, WC ≥ 95 cm.

Subjects with no hypertension (T), diabetes (D), and hyperlipidemia (L) were classified into TDL 0, while those with one or more were classified into TDL 1.

BMI model 1 was adjusted for age and sex. BMI model 2 was adjusted for age, sex, smoking, heavy drinking, physical activity, and low income. BMI model 3 was adjusted for age, sex, smoking, heavy drinking, physical activity, low income, DM, HTN, and hyperlipidemia. BMI model 4 was adjusted for age, sex, smoking, heavy drinking, physical activity, low income, DM, HTN, hyperlipidemia, COPD, and cancer. BMI model 5 was adjusted for age, sex, smoking, heavy drinking, physical activity, low income, DM, HTN, hyperlipidemia, COPD, cancer, and WC.

WC model 1 was adjusted for age and sex. WC model 2 was adjusted for age, sex, smoking, heavy drinking, physical activity, and low income. WC model 3 was adjusted for age, sex, smoking, heavy drinking, physical activity, low income, DM, HTN, and hyperlipidemia. WC model 4 was adjusted for age, sex, smoking, heavy drinking, physical activity, low income, DM, HTN, hyperlipidemia, COPD, and cancer. WC model 5 was adjusted for age, sex, smoking, heavy drinking, physical activity, low income, DM, HTN, hyperlipidemia, COPD, cancer, and BMI.

### 2.5. Anthropometry

Anthropometric measurements were taken during a physical examination by trained specialists. Height and weight units of measure were cm and kg, respectively. The measurements were adjusted to one decimal point (0.1 cm, 0.1 kg).

### 2.6. Definition of Variables

The health examination includes a self-administered questionnaire including questions pertaining to health-related life behavior, which is filled out by the examinee. According to the responses to questions in the questionnaire, heavy drinking was defined as daily average alcohol consumption of >30 g. Regular physical activity was defined as moderate-intensity exercise of more than 5 times a week or high-intensity exercise of more than 3 times a week. Low income was defined as being in the bottom 20% income group or having received [social aid].

### 2.7. Statistical Analysis

SAS version 9.4 (SAS Institute Inc., Cary, NC, USA.) was used for statistical analysis. The data were expressed as means ± SD, N (%). The differences between groups were adjusted using Cox’s proportional hazard model (hazard ratio [HR]; 95% confidence interval [CI]), analysis of variance (ANOVA) test, χ^2^ test with the study variables (sex, age, smoking, heavy drinking, low income, regular physical activity, hypertension, DM, and hyperlipidemia), and verified with categorical variable distribution differences. The cause-specific model was used for conducting a competitive analysis of death. *p*-values less than 0.05 were considered statistically significant.

## 3. Results

The general characteristics of participants are shown in Table 1 and Table 2. The average BMI of the participants was 23.1 ± 4.9 kg/m^2^ and the average WC was 82.2 ± 9.2 cm. The proportion of men was (39.25%) lower than that of women. The proportion of men was the highest in the BMI level 2 group, which is the normal weight group, and the proportion decreased from levels 3 to 5. The average age of the participants was the highest in the BMI level 1 group and decreased from levels 2 to 5. The proportion of men was the highest in the WC level 1 group and decreased from levels 2 to 5. The average age was the highest in the WC level 1 group and decreased from levels 2 to 4, but increased slightly in the level 5 group.

The results of correlation analysis between 5 BMI levels and MI incidence depending on adjustment variables using Cox’s proportional hazard model are shown in Table 3. The adjustment variables were age, sex, smoker, heavy drinker, regular physical activity, low income, diabetes, hypertension, hyperlipidemia, COPD, cancer, and WC, and the model was divided into five categories.

In analyzing the correlation between BMI levels and MI incidence, adjustment variables were sequentially added. As a result, for most statistically significant models, in the case of BMI level 1 (lowest BMI), MI incidence was significantly high for every model. In contrast, in the case of BMI levels 3, 4, and 5 (higher BMI) from model 3 which was adjusted with hypertension, diabetes, and others, MI incidence significantly decreased and it was consistent even after adding WC adjustment. As a result of the tendency test, MI incidence significantly decreased as BMI increased for models 3, 4, and 5.

The results of correlation analysis between five levels of WC and MI incidence depending on adjustment variables using Cox’s proportional hazard model are shown in Table 4. The adjustment variables were age, sex, smoking, heavy drinking, regular physical activity, low income, diabetes, hypertension, hyperlipidemia, COPD, cancer, and BMI, and the model was divided into five categories. In analyzing the correlation between WC levels and MI incidence, adjustment variables were sequentially added. As a result, for most statistically significant models, in the case of WC level 2, MI incidence decreased, and in the case of WC levels 1 and 5, MI incidence increased compared to the standard group. In addition, for model 4, in the case of WC levels 1 and 2, MI incidence was higher than that of the standard group. However, in model 5, which was adjusted for BMI, the MI incidence for WC level 2 was lower than that of the standard group. In contrast, the MI incidence for WC level 1 was higher than that of the standard group, but not statistically significant. Incidence of new-onset MI according to WC did not increase or decrease in one direction; therefore, a tendency test was not conducted.

Taken together, in the case of BMI, the HR for MI changed into a negative relationship with BMI in model 3, which was adjusted for hypertension, DM, and hyperlipidemia, and this was maintained up to model 5, which was further adjusted for WC. In contrast, in the case of WC, the HR for MI decreased in WC greater than the normal level in model 3, which was adjusted for hypertension, DM, and hyperlipidemia. In model 5, which was further adjusted for BMI, the HR for MI changed into a positive relationship with WC.

Table 5 shows the results of the analysis of the effect of hazards on MI incidence using model 5. Participants who are more than 85 years old, had low income, or had diabetes, hypertension, hyperlipidemia, or COPD showed a higher hazard ratio than the standard group; these were all statistically significant. Participants who are male, heavy drinkers, or had regular physical activity or cancer showed a lower hazard ratio than the standard group; all except cancer were statistically significant.

The analysis result by groups is as follows: Groups having one or more of hypertension, DM, or hyperlipidemia showed higher HR than groups without hypertension, DM, or hyperlipidemia in every BMI and WC group; moreover, the lowest HR group among those having one or more of hypertension, DM, or hyperlipidemia showed higher HR than the highest HR group among those having no hypertension, DM, or hyperlipidemia (Appendix A). Figure 1a is the graph representing hazard ratios for MI by groups for BMI, and Figure 1b, hazard ratios for MI by groups for WC. In both graphs, groups having one or more of hypertension, diabetes, or hyperlipidemia showed higher hazard ratios than the groups having no hypertension, diabetes, or hyperlipidemia; Figure 1a shows that as BMI increased, the hazard ratio for MI decreased. As a result of the tendency test, MI incidence significantly decreased as BMI increased on both TDL 0 and TDL 1. Figure 1b shows that as WC increased, the hazard ratio for MI increased. As a result of the tendency test, MI incidence significantly increased as WC increased on both TDL 0 and TDL 1.

The analysis result by BMI and WC segmented sections using model 5, which was adjusted for every variable, for more continuous verification is that as BMI increases, MI HR decreases, and as WC increases, MI HR increases (Appendix A). Figure 2a is the graph representing the hazard ratio for MI listed in Appendix A. It showed that as BMI increased, the hazard ratio for MI decreased. As a result of the tendency test, MI incidence significantly decreased as BMI increased. Figure 2b is the graph representing the hazard ratio for MI listed in Appendix A. It showed that as WC increased, the hazard ratio for MI also increased. As a result of the tendency test, MI incidence significantly increased as WC increased.

As a result, when using model 5, every analysis result showed that as BMI decreases, MI HR increases, and as BMI increases, MI HR decreases; concerning WC, lower than normal levels of the WC groups showed low MI HR or a high but statistically non-significant result and the abdominal obesity group showed increased HR.

## 4. Discussion

This study showed that when every variable, including WC, was adjusted, BMI was inversely related to HR for MI. WC-adjusted elevated BMI was attributed to an increased bone density, limb muscle mass, or subcutaneous fat. We also observed that when every variable, including BMI, was adjusted, HR for MI proportionally changes with WC. BMI-adjusted WC was attributed to increased abdominal fat and/or decreased bone density, limb muscle mass, or subcutaneous fat. ESRD, which may play a role in water retention, occurred in only 0.106% (in 805 out of 758,471 participants) of the participants (Table 1).

### 4.1. Limitation of BMI as a Screening Method for Obesity Diagnosis

The World Health Organization defined overweight and obesity as abnormal or excessive fat accumulation that may impair health [27]. The most ideal screening method for obesity diagnosis may be the obesity index, which represents the obesity level where health is impaired in all sex, age groups, and race; however, it does not exist presently [28].

Currently, BMI is widely used as a standard screening method for body fat measurement, but its inability to distinguish between lean body mass and body fat has limited its utility [29]. The use and interpretation of BMI may be even more limited in older adults who may have a loss of muscle with aging and increased visceral fat due to the redistribution of adipose tissue [30].

Height is one of two factors to derive BMI. Although it differs depending on sex, height has problems not only with reduction, which generally happens after 40 years (2–3 cm height reduction/decade), caused by age-related physiologic changes, such as spinal deformity, thinning of the intervertebral discs [31] but also measurement difficulty generated from pathological changes with aging such as akyphosis, scoliosis, or bent legs, which makes meeting the height measurement standard difficult as it may be difficult for parts of the body (heels, back, hip, and back of the head) to touch a perpendicular plate [32].

### 4.2. Obesity Paradox

The obesity paradox remains controversial at present. Many studies refute it, citing bias; many speculate about its value in association with BMI and CVD mortality and prognosis. Kang et al. [33] researched 3824 patients with STEMI treated with PCI within 12 h of chest pain onset. The research was conducted based on the BMI of four groups, and prognosis in terms of death while hospitalized, revascularization within a year and death within a year were compared; as a result, overweight and obesity groups showed better short- and long-term prognosis than the underweight group. The researchers speculated that the reasons were that overweight and obesity patients were regarded as a risk group of CVD in advance; therefore, medicines such as a statin, aspirin, beta blocker, renin-angiotensin inhibitor, and hyperlipidemic agents were more proactively prescribed, and the fact that the average age of the overweight and obesity groups was 10–13 years younger than the underweight group acted as bias. In addition, because obesity takes a long time to affect the incidence of various diseases, the follow-up period was too short to identify the effect. Hӓllberg et al. [34] studied 922 patients treated with CABG in a 20-year follow-up study and reported that death from CVD was higher in the normal weight group than in the overweight and obesity groups 10 years after surgery; however, death from CVD was higher in the overweight and obese groups than in the normal participants at the end of 20-year follow-up after CABG.

Many studies speculate that the obesity paradox is a result of contradicting interpretations of theories surrounding weight and its associated health issues [35]. Smoking has been known to decrease weight. However, many studies reported it as a confounder due to the crude classification and incorrect estimation; moreover, this disturbance occurred even in the studies adjusted for smoking [36], and this argument was based on studies concerning the association between the smoking habits survey and actual blood cotinine concentrations representing only 0.40–0.70 [37].

In the present study, analysis of the correlation between obesity indicators and HR for MI showed that WC was positively correlated with HR for MI in those with abdominal obesity, whereas BMI was negatively correlated. In other words, the obesity paradox that overweight or obese people show better clinical outcomes appeared to be applicable to BMI, but not to abdominal obesity. Such findings showed that the obesity paradox reflects differences in results according to the measurement method for obesity, while also demonstrating that the obesity paradox is not applicable to abdominal obesity.

### 4.3. WC as a Screening Parameter for Obesity

Several epidemiological studies demonstrated that WC is more strongly associated with various obesity-related diseases. Sobiczewski et al. [38] reported that WC was positively correlated with all-cause mortality and cardiovascular mortality, whereas BMI showed negative correlations.

In addition, there have been studies that explained the mechanism by which body fat, especially abdominal and visceral fat, caused MI, DM, hypertension, and hyperlipidemia. Matsuzawa et al. [39] demonstrated that adipogenesis occurred more actively in mesenteric fat than in subcutaneous fat based on the observation of enhanced mRNA synthesis of acyl-CoA, a key enzyme for adipogenesis. These findings suggested that visceral fat is metabolically active, and free fatty acids can be released directly into the liver through the portal circulation, and as a result, an excessive amount of free fatty acids accumulate in the liver, promoting lipid synthesis in the liver, inducing insulin resistance that causes hyperlipidemia, glucose tolerance, and hypertension and ultimately leading to arteriosclerosis.

Other studies have also reported that excessive fat accumulation could cause chronic inflammation, which acts as a mechanism for obesity-related metabolic disorders and macrovascular complications. The evidence that the immune system is activated in obese patients is supported by increased levels of pro-inflammatory cytokines in the blood and infiltration of macrophages and other immune cells into adipose tissues, liver, muscles, and pancreas. As the production of pro-inflammatory cytokines increases, immune cells switch to a pro-inflammatory state. This eventually causes insulin secretion deficiency via insulin signaling in peripheral tissues and beta-cell dysfunction [40]. Moreover, C-reactive protein (CRP) secretion is regulated in the liver by pro-inflammatory cytokines, such as IL-6 and TNF-α. Meanwhile, the CRP level is associated with insulin resistance, hypertension, high-density cholesterol, triglycerides, and vascular endothelial dysfunction [41]. It can ultimately activate blood clotting mechanisms that cause vascular thrombotic disorders [42]. Furthermore, visceral fat is more metabolically active than peripheral fat; thus, visceral fat can also be considered the source of low-grade chronic inflammation that causes atherosclerosis [43].

### 4.4. Aging, Sarcopenia, and Sarcopenic Obesity

Muscle mass decreases by approximately 30% between the ages of 20 and 80 years [44]. The causes of muscle loss include mitochondrial dysfunction in muscle cells, oxidative stress, neurological changes, hormonal changes, nutritional deficiencies, decreased physical activities, and low-grade inflammation [45]. In addition to decreased muscle mass, an increase in visceral fat and intermuscular fat can occur, resulting in metabolic disorders related to insulin resistance and inflammation of adipose tissues [46]. Decreased muscle mass also acts as a risk factor for CVD by causing falls, general weakness, and decreased physical activity [47]. It can also cause a reduced basal metabolic rate that leads to an imbalance between energy intake and consumption, whereby unconsumed excess energy is stored in fatty tissues, especially visceral fat [48,49]. In addition to their original functions, skeletal muscles represent the largest organ system in the body that accounts for 40–50% of total body weight, and they not only consume energy but also act as metabolic organs involved in insulin sensitivity and protein synthesis. Accordingly, decreased muscle mass itself could be a risk factor for CVD [50]. Various epidemiological studies have reported consistent findings on increased incidence and total mortality rate associated with metabolic syndrome and CVD in patients with sarcopenic obesity [51,52,53,54,55,56]. Many studies have reported that sarcopenia alone can increase the incidence and total mortality rate associated with metabolic syndrome and CVD [52,53,57,58,59]. Other studies showed that the incidence and total mortality rate significantly increased only in patients with both sarcopenia and obesity [56].

Not only decreased muscle mass but also aging-related loss of muscle strength can increase the risk of functional deterioration in older adults [60], while also causing disability, death, and other adverse health effects [61]. Lazarus et al. [62] reported that skeletal muscle strength measured by grip strength is negatively correlated with fasting insulin level. Meanwhile, Jurca et al. [63] reported that muscle strength and metabolic syndrome have an independent negative correlation with each other. Reduced muscle strength could be attributed to decreased muscle mass on one hand. On the other hand, it can be explained as decreased muscle function, meaning a decrease in force that can be generated per unit area of skeletal muscle. The mechanisms of such reduced muscle strength associated with aging include changes in cellular or molecular processes such as decreased concentration or ability to regenerate, the interaction of muscle filaments, decreased mitochondrial function, and increased adipocytes in skeletal muscles [64,65].

### 4.5. Aging and Changes in Fat Distribution

Aging not only leads to changes in the total fat content in the body but also to the redistribution of fat. As a result, abdominal fat, especially visceral fat, tends to increase, whereas subcutaneous fat in the lower extremities tends to decrease [66]. As previously described, several studies showed that abdominal obesity causes arteriosclerosis and metabolic syndrome, thereby increasing the incidence of CVD, whereas other studies have reported that, unlike visceral fat, peripheral fat has anti-arteriosclerotic effects. In a cohort study by Tankὀ et al. [67] on 1356 Danish women aged 60 to 85 years, the findings showed that blood lipid levels that cause arteriosclerosis were lowest in the group with a low percentage of visceral fat and a high percentage of peripheral fat and that peripheral fat had independently excellent anti-arteriosclerotic effects.

The ability to store fat in the subcutaneous tissues, especially in the lower extremities, gradually declines with aging. Consequently, adipose cells lose the ability to function as a fat reservoir or metabolic sinks, preventing adequate absorption and buffering of free fatty acids in blood [68]. This also causes an increase in the concentration of free fatty acids in blood and such excess fat may accumulate as visceral, liver, or intramuscular fat [69]. Moreover, subcutaneous adipose tissue acts not only as an organ that functions as an energy reservoir and insulator but also as an endocrine or paracrine organ that secretes hormones such as adiponectin to affect the cardiovascular system. Adiponectin is a type of adipokine secreted only by white adipocytes and is responsible for key mechanisms for alleviating the pathological mechanism of CVD based on its anti-inflammatory, anti-oxidant, and anti-apoptotic effects [70]. Because adiponectin can increase insulin sensitivity, it is also known to prevent obesity-related metabolic disorders [71], whereas low blood concentration of adiponectin is believed to be associated with DM and CVD [72].

Among studies that explored the association between adiponectin secretion and distribution of body fat, a study by Guenther et al. [73] on 424 Caucasian men and women reported that adiponectin concentration was higher among those with more subcutaneous fat than visceral fat. In addition, another study by Gariballa [74] on 206 overweight or obese patients reported that blood adiponectin levels decreased as visceral fat increased. Another study that supported such findings was an in vitro study by Reneau et al. [75], which cultured subcutaneous and visceral adipose tissues collected from 55 patients undergoing abdominal surgery and retrieved the media after 24 h to measure the adiponectin concentration by ELISA. The results showed that adiponectin concentration was higher in subcutaneous adipose tissues than in visceral adipose tissues. In an in vivo study on 2820 patients, measurement of body composition using dual-energy radiation absorptiometry and comparison of plasma adiponectin concentrations showed that adiponectin concentration increased as adipose tissues in the lower extremities increased and adiponectin concentration decreased as adipose tissues in the trunk increased, regardless of sex or race [76].

This study was the first to demonstrate that WC, rather than BMI, is a suitable obesity indicator for predicting the incidence of MI among the Korean elderly population aged 75 years or older. Moreover, this study also identified the effects of BMI and WC on MI as indicators of obesity among older adults aged 75 years or older, in whom sarcopenia, osteoporosis, and accumulation of visceral fat can be more preponderant than in other age groups. Lastly, this study explored the theoretical feasibility of using BMI and WC to predict the risk of cardiovascular disease.

This study had the following limitations. The first limitation is information bias associated with using BMI. Although BMI is a reliable indicator when anthropometric measurements are made accurately, BMI may be distorted due to the fact that height measurements according to the standard are difficult for the elderly as their height may have decreased and their back and legs may have become curved due to aging. The second limitation is selection bias. The study included older adults aged 75 years or older, most of whom are retirees with a great interest in their own health to voluntarily participate in a national health screening. However, individuals who are unable to leave their homes for examination due to general weakness, serious disability, or serious chronic disease were excluded from the cohort of this study. Additionally, just as in other previous studies on the elderly, the surviving population with relatively good health would have been included in the cohort. Thirdly, this study used health screening data from the National Health Insurance Service, unlike clinical trials that investigate the participants according to schedule. As a result, changes in the body weight of the participants until the completion of the study could not be observed. Fourthly, we did not exclude comorbidities such as DM, HTN, hyperlipidemia, and cancer initially. To conduct the analysis of the general population, we tried to overcome this limitation by adjusting for the comorbidities rather than their exclusion. It is hoped that future follow-up studies can overcome such limitations.

## 5. Conclusions

This study analyzed the effects of BMI and WC on the incidence of MI in the Korean population aged over 75 years. This study’s results showed that HR for MI has a negative relationship with BMI, whereas it has a positive relationship with WC. Furthermore, WC is a more appropriate indicator for predicting the risk of MI in the older population. Further, this study also verified that the obesity paradox only applies to BMI, not to WC. Therefore, the findings of this study may contribute toward establishing proper guidelines for obesity management among the elderly aged 75 years or older. In conclusion, this study recommends focusing on reducing abdominal obesity, rather than weight, and increasing limb muscle mass in preventing MI in the older adult population.

## Figures and Tables

**Figure 1 medicina-58-01768-f001:**
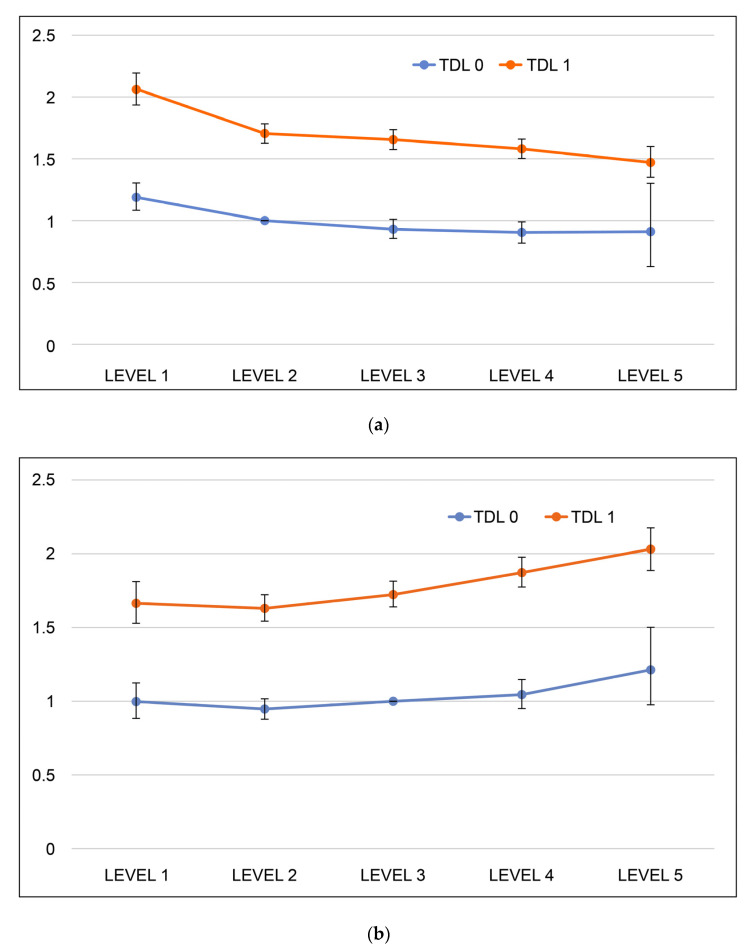
Graph representing hazard ratio for MI by groups listed in Appendix A. (**a**) *p* for trend TDL 0 = 0.025/TDL 1 = 0.007, (**b**) *p* for trend TDL 0 = 0.025/TDL 1 = 0.025. T, hypertension; D, diabetes; L, hyperlipidemia.

**Figure 2 medicina-58-01768-f002:**
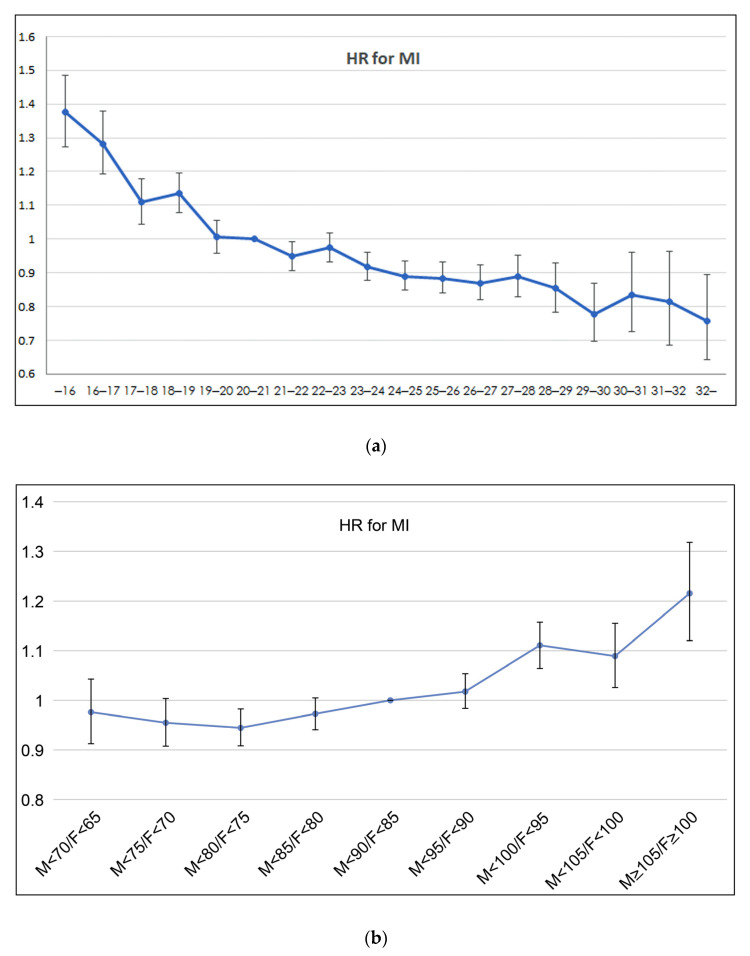
Graph representing hazard ratio for MI listed in Appendix A. (**a**) *p* for trend < 0.001, (**b**) *p* for trend = 0.003. HR, hazard ratio; MI, myocardial infarction.

**Table 1 medicina-58-01768-t001:** General characteristics of participants.

Based on BMI						
	BMI Levels					
	1	2	3	4	5	*p*-Value
Number (%)	56,693 (7.5)	313,840 (41.4)	174,820 (23.0)	193,677 (25.5)	19,441 (2.6)	
Sex, Male (39.25%)	23,756 (41.9%)	136,366 (43.5%)	71,170 (40.7%)	63,188 (32.6%)	3196 (16.4%)	<0.001
WC (cm)	70.26 ± 7.0	77.94 ± 6.8	84.15 ± 6.3	89.47 ± 6.9	97.28 ± 8.2	<0.001
Age (years)	80.86 ± 4.8	79.56 ± 4.0	78.91 ± 3.5	78.63 ± 3.3	78.44 ± 3.2	<0.001
Height (cm)	153.79 ± 9.8	154.29 ± 9.5	154.39 ± 9.3	153.42 ± 8.9	150.45 ± 8.4	<0.001
Weight (kg)	40.76 ± 5.9	50.26 ± 7.0	57.21 ± 7.1	63.04 ± 7.7	72.18 ± 8.6	<0.001
SBP (mmHg)	126.36 ± 17.9	130.1 ± 17.1	132.35 ± 16.6	133.89 ± 16.4	135.87 ± 16.7	<0.001
DBP (mmHg)	76.27 ± 10.8	77.48 ± 10.4	78.24 ± 10.3	79.01 ± 10.2	80.33 ± 10.4	<0.001
Total cholesterol (mg/dL)	186.47 ± 43.5	193.85 ± 45.2	197.71 ± 44.5	199.78 ± 45.9	202.99 ± 50.9	<0.001
Fasting glucose (mg/dL)	99.41 ± 26.8	102.14 ± 27.8	104.86 ± 28.6	107.03 ± 29.2	110.98 ± 32.0	<0.001
Smoker						<0.001
Non	40,686 (71.8)	232,798 (74.2)	135,258 (77.4)	158,951 (82.1)	17,416 (89.6)	
Ex	7108 (12.5)	44,319 (14.1)	25,474 (14.6)	23,961 (12.4)	1422 (7.3)	
Current	8899 (15.7)	36,723 (11.7)	14,088 (8.1)	10,765 (5.6)	603 (3.1)	
Heavy drinker	1727 (3.1)	11,220 (3.6)	5166 (3.0)	4376 (2.3)	240 (1.2)	<0.001
Regular physical activity	4616 (8.1)	38,271 (12.2)	24,922 (14.3)	25,913 (13.4)	1878 (9.7)	<0.001
Low income (<20%)	12,527 (22.1)	64,883 (20.7)	35,009 (20.0)	38,940 (20.1)	4123 (21.2)	<0.001
Hypertension	31,741 (56.0)	208,922 (66.6)	132,801 (76.0)	160,493 (82.9)	17,473 (89.9)	<0.001
Diabetes	11,609 (20.5)	78,995 (25.2)	54,121 (31.0)	69,307 (35.8)	8438 (43.4)	<0.001
Hyperlipidemia	12,308 (21.7)	99,982 (31.9)	70,757 (40.5)	90,050 (46.5)	10,107 (52.0)	<0.001
COPD	12,350 (21.8)	55,237 (17.6)	29,315 (16.8)	32,095 (16.6)	3402 (17.5)	<0.001
ESRD	63 (0.1)	381 (0.1)	184 (0.1)	167 (0.1)	10 (0.1)	0.001
Cancer	3128 (5.5)	16,334 (5.2)	8479 (4.9)	8500 (4.4)	676 (3.5)	<0.001
WC levels						<0.001
1	17,647 (31.1)	10,891 (3.5)	503 (0.3)	301 (0.2)	44 (0.2)	
2	28,984 (51.1)	121,135 (38.6)	13,051 (7.5)	2549 (1.3)	158 (0.8)	
3	8829 (15.6)	154,815 (49.3)	104,748 (59.9)	54,381 (28.1)	834 (4.3)	
4	1144 (2.02)	25,578 (8.2)	53,044 (30.3)	108,545 (56.0)	6408 (33.0)	
5	89 (0.2)	1421 (0.5)	3474 (2.0)	27,901 (14.4)	11,997 (61.7)	
**According to WC**						
	**WC Levels**					***p*-Value**
	**1**	**2**	**3**	**4**	**5**	
Number (%)	29,386 (3.9)	165,877 (21.9)	323,607 (42.7)	194,719 (25.7)	44,882 (5.9)	
Sex, Male (39.25%)	15,277 (52.0%)	76,739 (46.3%)	131,264 (40.6%)	64,166 (33.0%)	10,230 (22.8%)	<0.001
BMI	18.17 ± 2.2	20.31 ± 2.1	22.9 ± 2.3	25.47 ± 7.5	28.32 ± 3.0	<0.001
Age (years)	80.54 ± 4.8	79.63 ± 4.1	79.14 ± 3.7	78.94 ± 3.5	78.99 ± 3.5	<0.001
Height (cm)	153.9 ± 9.6	153.75 ± 9.4	154.06 ± 9.4	154.09 ± 9.3	153.44 ± 8.9	<0.001
Weight (kg)	43.12 ± 7.0	48.15 ± 7.4	54.52 ± 8.2	60.6 ± 8.9	66.82 ± 10.0	<0.001
SBP (mmHg)	125.23 ± 17.6	128.88 ± 17.2	131.64 ± 16.8	133.47 ± 16.6	134.98 ± 16.8	<0.001
DBP (mmHg)	75.74 ± 10.5	77.08 ± 10.4	78.04 ± 10.3	78.78 ± 10.3	79.65 ± 10.6	<0.001
Total cholesterol (mg/dL)	183.66 ± 39.6	190.91 ± 43.1	196.61 ± 45.4	199.49 ± 45.8	202.26 ± 51.7	<0.001
Fasting glucose (mg/dL)	98.52 ± 25.5	99.94 ± 25.2	103.69 ± 28.0	107.26 ± 30.3	111.29 ± 33.5	<0.001
Smoker						<0.001
Non	20,370 (69.3)	120,706 (72.8)	248,566 (76.8)	157,176 (80.7)	38,291 (85.3)	
Ex	4257 (14.5)	23,304 (14.1)	45,244 (14.0)	24,917 (12.8)	4562 (10.2)	
Current	4759 (16.2)	21,867 (13.2)	29,797 (9.2)	12,626 (6.5)	2029 (4.5)	
Heavy drinker	872 (3.0)	5948 (3.6)	10,115 (3.1)	4958 (2.6)	836 (1.9)	<0.001
Regular physical activity	2818 (9.6)	19,946 (12.0)	43,595 (13.5)	24,718 (12.7)	4523 (10.1)	<0.001
Low income (<20%)	6117 (20.8)	34,720 (20.9)	65,622 (20.3)	39,725 (20.4)	9298 (20.7)	<0.001
Hypertension	15,495 (52.7)	101,946 (61.5)	235,822 (72.9)	158,733 (81.5)	39,434 (87.9)	<0.001
Diabetes	5552 (18.9)	34,939 (21.1)	92,269 (28.5)	70,225 (36.1)	19,485 (43.4)	<0.001
Hyperlipidemia	5841 (19.9)	45,537 (27.5)	120,883 (37.4)	88,077 (45.2)	22,866 (51.0)	<0.001
COPD	6067 (20.7)	29,662 (17.9)	54,884 (17.0)	33,558 (17.2)	8228 (18.3)	<0.001
ESRD	36 (0.1)	196 (0.1)	330 (0.1)	197 (0.1)	46 (0.1)	0.4
Cancer	1713 (5.8)	8868 (5.4)	15,865 (4.9)	8885 (4.6)	1786 (4.0)	<0.001
BMI levels						<0.001
1	17,647 (60.1)	28,984 (17.5)	8829 (2.7)	1144 (0.6)	89 (0.2)	
2	10,891 (37.1)	121,135 (73.0)	154,815 (47.8)	25,578 (13.1)	1421 (3.2)	
3	503 (1.7)	13,051 (7.9)	104,748 (32.4)	53,044 (27.2)	3474 (7.7)	
4	301 (1.0)	2549 (1.5)	54,381 (16.8)	108,545 (55.7)	27,901 (62.2)	
5	44 (0.2)	158 (0.1)	834 (0.3)	6408 (3.3)	11,997 (26.7)	

*p*-values were obtained by analysis of variance (ANOVA) test. Data are presented as means ± SD (Standard Deviation) or N (%). BMI, body mass index; SBP, systolic blood pressure; DBP, diastolic blood pressure; COPD, chronic obstructive pulmonary disease; ESRD, end-stage renal disease; MI, myocardial infarction; WC, waist circumference.

**Table 2 medicina-58-01768-t002:** General characteristics and risk factors in accordance with new-onset MI.

	MI		*p*-Value
	No	Yes	
**N (%)**	**724,049 (95.5)**	**34,422 (4.5)**	
Sex, Male	282,915 (39.1)	14,761 (42.9)	<0.001
Body mass index (kg/m^2^)	23.1 ± 5.0	23.2 ± 3.3	0.002
Waist circumference (cm)	82.2 ± 9.2	82.9 ± 9.0	
Age (years)	79.2 ± 3.8	79.4 ± 3.7	<0.001
Height (cm)	153.9 ± 9.3	154.3 ± 9.4	<0.001
Weight (kg)	55.0 ± 10.0	55.4 ± 10.1	<0.001
SBP (mmHg)	131.4 ± 17.0	132.6 ± 17.4	<0.001
DBP (mmHg)	78.0 ± 10.4	78.6 ± 10.7	<0.001
Total cholesterol (mg/dL)	195.8 ± 45.3	199.3 ± 47.6	<0.001
Blood glucose (mg/dL)	103.9 ± 28.2	106.7 ± 33.1	<0.001
Smoker			<0.001
Non	559,806 (77.3)	25,303 (73.5)	
Ex	97,394 (13.5)	4890 (14.2)	
Current	66,849 (9.2)	4229 (12.3)	
Heavy drinker	21,680 (3.0)	1049 (3.1)	0.572
Regular physical activity	91,626 (12.7)	3974 (11.6)	<0.001
Low income (<20%)	148,289 (20.5)	7193 (20.9)	0.062
Hypertension	524,037 (72.4)	27,393 (79.6)	<0.001
Diabetes	209,815 (29.0)	12,655 (36.8)	<0.001
Hyperlipidemia	267,704 (37.0)	15,500 (45.0)	<0.001
COPD	125,004 (17.3)	7395 (21.5)	<0.001
ESRD	736 (0.1)	69 (0.2)	<0.001
Cancer	35,691 (4.9)	1426 (4.1)	<0.001
BMI levels			<0.001
1	54,278 (7.5)	2415 (7.0)	
2	299,957 (41.4)	13,883 (40.3)	
3	166,587 (23.0)	8233 (23.9)	
4	184,636 (25.5)	9041 (26.3)	
5	18,591 (2.6)	850 (2.5)	
WC levels			<0.001
1	28,217 (3.9)	1169 (3.4)	
2	158,861 (21.9)	7016 (20.4)	
3	308,862 (42.7)	14,745 (42.8)	
4	185,390 (25.6)	9329 (27.1)	
5	42,719 (5.9)	2163 (6.3)	

*p*-values were obtained by χ2 test. Data are presented as means ± SD (standard deviation) or N (%). BMI, body mass index; SBP, systolic blood pressure; DBP, diastolic blood pressure; COPD, chronic obstructive pulmonary disease; ESRD, end-stage renal disease; MI, myocardial infarction; WC, waist circumference.

**Table 3 medicina-58-01768-t003:** Cox’s proportional hazard model for the incidence of new-onset MI according to BMI.

BMI Levels	N	MI	Model 1	Model 2	Model 3	Model 4	Model 5
1	56,693	2415	1.132 (1.084,1.182)	1.117 (1.070,1.166)	1.222 (1.17,1.277)	1.211 (1.159,1.265)	1.235 (1.180,1.291)
2	313,840	13,883	1 (Ref.)	1 (Ref.)	1 (Ref.)	1 (Ref.)	1 (Ref.)
3	174,820	8233	1.025 (0.997,1.053)	1.032 (1.004,1.061)	0.958 (0.932,0.985)	0.960 (0.934,0.986)	0.945 (0.918,0.973)
4	193,677	9041	1.025 (0.998,1.052)	1.034 (1.006,1.062)	0.912 (0.888,0.937)	0.913 (0.889,0.939)	0.887 (0.859,0.916)
5	19,441	850	1.013 (0.945,1.086)	1.018 (0.950,1.092)	0.851 (0.793,0.912)	0.848 (0.791,0.910)	0.806 (0.748,0.870)
*p* for trend			0.381	0.954	<0.001	<0.001	<0.001

Data are presented as hazard ratio (95% CI). Model 1: adjusted for age and sex. Model 2: adjusted for age, sex, smoking, heavy drinking, physical activity, and low income. Model 3: adjusted for age, sex, smoking, heavy drinking, physical activity, low income, DM, HTN, and hyperlipidemia. Model 4: adjusted for age, sex, smoking, heavy drinking, physical activity, low income, DM, HTN, hyperlipidemia, COPD, and cancer. Model 5: adjusted for age, sex, smoking, heavy drinking, physical activity, low income, DM, HTN, hyperlipidemia, COPD, cancer, and WC. BMI, body mass index; MI, myocardial infarction; DM, diabetes mellitus; HTN, hypertension; COPD, chronic obstructive pulmonary disease.

**Table 4 medicina-58-01768-t004:** Cox’s proportional hazard model for the incidence of new-onset MI according to WC.

WC Levels	N	MI	Model 1	Model 2	Model 3	Model 4	Model 5
1	29,386	1169	1.006 (0.948,1.068)	0.991 (0.933,1.052)	1.151 (1.084,1.222)	1.145 (1.078,1.216)	1.002 (0.940,1.067)
2	165,877	7016	0.962 (0.935,0.989)	0.955 (0.929,0.983)	1.043 (1.013,1.073)	1.041 (1.012,1.072)	0.968 (0.938,0.998)
3	323,607	14,745	1 (Ref.)	1 (Ref.)	1 (Ref.)	1 (Ref.)	1 (Ref.)
4	194,719	9329	1.055 (1.028,1.083)	1.055 (1.028,1.083)	0.984 (0.959,1.011)	0.983 (0.958,1.009)	1.058 (1.028,1.088)
5	44,882	2163	1.098 (1.050,1.149)	1.096 (1.047,1.147)	0.969 (0.926,1.014)	0.964 (0.921,1.009)	1.127 (1.070,1.187)

Data are presented as hazard ratio (95% CI). Model 1: adjusted for age and sex. Model 2: adjusted for age, sex, smoking, heavy drinking, physical activity, and low income. Model 3: adjusted for age, sex, smoking, heavy drinking, physical activity, low income, DM, HTN, and hyperlipidemia. Model 4: adjusted for age, sex, smoking, heavy drinking, physical activity, low income, DM, HTN, hyperlipidemia, COPD, and cancer. Model 5: adjusted for age, sex, smoking, heavy drinking, physical activity, low income, DM, HTN, hyperlipidemia, COPD, cancer, and BMI. MI, myocardial infarction; WC, waist circumference; BMI, body mass index; DM, diabetes mellitus; HTN, hypertension; COPD, chronic obstructive pulmonary disease.

**Table 5 medicina-58-01768-t005:** Multivariate hazard ratio for MI incidence using model 5 (Cox’s proportional hazard model).

		Hazard Ratio (95% CI)	*p*-Value
Age	≥85	1.044 (1.041, 1.047)	<0.001
	<85	1.000 (ref)	
Sex	Male	0.792 (0.771, 0.814)	<0.001
	Female	1.000 (ref)	
Smoker	yes	1.195 (1.161, 1.230)	<0.001
	no	1.000 (ref)	
Heavy drinker	yes	0.932 (0.875, 0.992)	0.028
	no	1.000 (ref)	
Regular physical activity	yes	0.820 (0.793, 0.848)	<0.001
	no	1.000 (ref)	
Low income (<20%)	yes	1.061 (1.034, 1.089)	<0.001
	no	1.000 (ref)	
Diabetes	yes	1.386 (1.355, 1.418)	<0.001
	no	1.000 (ref)	
Hypertension	yes	1.407 (1.369, 1.445)	<0.001
	no	1.000 (ref)	
Hyperlipidemia	yes	1.289 (1.260, 1.318)	<0.001
	no	1.000 (ref)	
COPD	yes	1.321 (1.287, 1.355)	<0.001
	no	1.000 (ref)	
Cancer	yes	0.994 (0.942, 1.048)	0.823
	no	1.000 (ref)	

Adjusted for age, sex, smoking, heavy drinking, physical activity, low income, DM, hypertension, hyperlipidemia, COPD, and cancer. BMI (WC): Corresponding variables are excluded. BMI, body mass index; COPD, chronic obstructive pulmonary disease; MI, myocardial infarction; WC, waist circumference.

## Data Availability

The data analyzed or generated from this study are available at Korea NHIS (national health insurance system)—Senior database (NHISS: https://nhiss.nhis.or.kr/) (accessed on 28 September 2022).

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
