# Peer review of "Associations between Body Mass Index, Waist Circumference, and Myocardial Infarction in Older Adults Aged over 75 Years: A Population-Based Cohort Study"

_medicina, 2022, doi:10.3390/medicina58121768_

Round 1

Reviewer 1 Report

Interesting study inquiring the prognostic value of BMI and waist circumference in patients with advanced age.  The population-based design is attractive and the potential findings of general usefulness.

However, there are several issues to be clarified:

-) at line 73 authors say to have identified 758471 patients with more than 60 years, but below report that this number referred to over 75. please clarify

-) the use of code for the diagnosis of course is a limitation about the ascertainement of comorbidities and also adverse events.

-) the inclusion of cancer and other comorbidities in the model of course would have affected the parameters under evaluation (for example cancer), please address this issue in the discussion and limitation, according to the metabolic changes occurring in those conditions.

-) please explain on which base the 2nd BMI cathegory and 3rd WC catherogy have been selected as reference for the other, in a cathegorical analusis instead perform analysis of the continuum with linear regression, avoiding the limitation of the cathegorization, especially because groups are not balanced in numbers. Similar observation if for table 6-1 and 6-2. About the table 6-2, the second cathergory listed, for example, include male patients with WC between 75 and 70 or all patients below 75, as now stated?

-) about the findings depicted in figure 1-2 and 2-2, have been perfomed trend and/or interaction analysis? if not, please condiser them.

-) table 5-1 and 5-1 are duplicate of the figures, please put them in supplementary.  similar consideration for fig 2-1 and 2-2

-) did the authors conduce competitive analysis for death)

-) have authors performed multiple testing correction?

-) please consider to remove the exact number in reporting cathegorical variables, to improve the readibility.

Author Response

Response to Reviewer 1 Comments

Point 1: at line 73 authors say to have identified 758471 patients with more than 60 years, but below report that this number referred to over 75. please clarify.

Response 1: Firstly, thank you for your time to review our manuscript.

We obtained the clinical data of 758,471 older adults ‘aged ≥60 years’ from the NHIS-Senior database. The phrase, ‘aged ≥60 years’ was incorrectly inserted in the translation process. We apologize for this error. Thank you for your detailed review and for pointing this out. We have corrected the error by deleting that part. 

Point 2: the use of code for the diagnosis of course is a limitation about the ascertainement of comorbidities and also adverse events.

Response 2: Thank you for your valuable comment. We agree that using medical records would have been better. As described in the methods section, we classified the population according to international standards. If this classification was not possible due to the limitation from the data’s characteristics, we applied the classification by considering the characteristics of the data. Please consider the following explanation. The National Health Insurance of Korea covers 97% of the entire population. If a participant had comorbidities relevant for MI (DM, HTN, hyperlipidemia, etc.) and was diagnosed with those comorbidities by a doctor, it would have been recorded in the NHIS. Therefore, we consider that this limitation would have a limited effect on our research. Further, model 5 was adjusted for major relevant comorbidities.

Point 3: the inclusion of cancer and other comorbidities in the model of course would have affected the parameters under evaluation (for example cancer), please address this issue in the discussion and limitation, according to the metabolic changes occurring in those conditions.

Response 3: Thank you for your pertinent suggestion. We understand your point of view. As widely known, the most powerful risk factors affecting the incidence of MI are DM, HTN, and hyperlipidemia. We divided the population into two groups according to the presence of one or more of these chronic conditions (Tables 5-1 and 5-2). In this analysis, we adjusted for comorbidities such as cancer. The results show that the HR for MI has a negative relationship with BMI but a positive relationship with WC, regardless of the presence of the above-mentioned conditions. We agree that we have to consider the chronic diseases to minimize the effect. As you mentioned, we performed the analysis without classifing these comorbidities, which can be a limitation of this study. We have added this point in the limitation section. Nonetheless, please consider the following explanation. The most important conclusion of this research is presented in Supplementary Tables 1-1, 1-2, 2-1, and 2-2, which was obtained using model 5, which was adjusted for cancer as well as age, sex, smoking, heavy drinking, physical activity, low income, DM, HTN, hyperlipidemia, COPD, and BMI (or WC).   

Point 4 : please explain on which base the 2nd BMI cathegory and 3rd WC catherogy have been selected as reference for the other, in a cathegorical analusis instead perform analysis of the continuum with linear regression, avoiding the limitation of the cathegorization, especially because groups are not balanced in numbers. Similar observation if for table 6-1 and 6-2. About the table 6-2, the second cathergory listed, for example, include male patients with WC between 75 and 70 or all patients below 75, as now stated?

Response 4: Thank you for your insightful questions. As you mentioned, it is appropriate to present measured data as continuous variables, but in this study, patient conditions were categorized according to international standards for comparability with other studies. In that process, some categories were recategorized for stable statistical estimation. As you are well aware, statistical estimation is impossible if the N corresponding to the category is too small or absent. As described in section 2.4 subject classification, unlike WHO, in the Asia-Pacific region and according to the Korean Society for the Study of Obesity, BMI < 18.5 kg/m2 is defined as underweight, 18.5 kg/ m2 ≤ BMI < 23 kg/ m2 as normal, 23 kg/m2 ≤ BMI < 25 kg/ m2 as pre-obesity, 25 kg/ m2 ≤ BMI < 30 kg/m2 as obesity stage 1, 30~34.9 kg/m2 as obesity stage 2, and ≥35 kg/m2 as obesity stage 3. As shown in Table 1-1, even though we set the threshold of BMI as ≥30 kg/m2, it only occurred in 2.6% of the population, which means that BMI ≥35 kg/m2 is very rare in Asians, especially in the old age. Therefore, we did not classify obesity into 3 stages but included participants with a BMI ≥30 kg/m2 in the same group and considered the normal BMI (level 2) as the reference. In case of WC, the normal range differs according to race and sex. The Korean Society for the Study of Obesity defines WC in the normal range for males as WC < 90 cm and females as WC < 85 cm. Unlike BMI, WC does not have a commonly applied range for being underweight, normal weight, or obese. Therefore, we classified WC into 5 levels based on the upper limit of normal range (males 90 cm and females 85 cm) in 10 cm increments and selected level 3, which is the normal upper limit, as the reference. According to your suggestion, we have added these sentences in section 2.4 subject classification.

We have simplified the notations in Supplementary Tables 2-1 and 2-2. Table 2-1 started with BMI< 16 kg/m2, 16 kg/m2 ≤ BMI < 17 kg/m2, and ends with 32 kg/m2 ≤ BMI. Similarly, Supplementary Table 2-2 starts with WC in males<70 cm /females<65 cm, 70 cm ≤ males < 75 cm/ 65 cm ≤ females < 70, and ends with males≥105 cm /females≥100 cm. We apologize for the confusion, and we have changed the notations in the tables to be clearer.

Point 5 : about the findings depicted in figure 1-2 and 2-2, have been perfomed trend and/or interaction analysis? if not, please condiser them.

Response 5: We have conducted the analysis for determining the p values for the trend and added them in the figures.

Point 6 : table 5-1 and 5-1 are duplicate of the figures, please put them in supplementary.  similar consideration for fig 2-1 and 2-2

Response 6: Thank you for pointing this out. We can move the tables to an appendix or include them as a supplementary file.

Point 7 : did the authors conduce competitive analysis for death?

Response 7: There are several statistical methods for establishing a competitive risk model, such as the cause-specifie model and Fine & Gray model. In this study, we conducted a competitive analysis for death using the cause-specific model. We appreciate your insight, and have added this information in section 2.8 statistical analysis.   

Point 8 : have authors performed multiple testing correction?

Response 8: As you mentioned, we performed multiple testing corrections in preparation for statistical analysis. However, most of the p-values were <0.0001; therefore, we proceeded without considering the multiple testing corrections.

Point 9 : please consider to remove the exact number in reporting cathegorical variables, to improve the readibility..

Response 9: We apologize for we do not fully understand your intended meaning. Can you please explain more specifically?

Lastly, we appreciate your time and efforts to review our paper and improve this manuscript.

Reviewer 2 Report

I would like to congratulate you on the number of results that the work contains, however:

1.    I'm a bit confused about the results...table 1.1 shows BMI levels...............but those aren't BMI levels, are they? Subsequently, the first line states number....numbers of what? A number of participants in the correct group? 56,69????,....sex, male?23,756? What is it?

Please explain the individual items in the table more clearly... similarly, table 1-2 and others.

line 289, why is there under the table abbreviation: WC?

2.    The weakest part of the article is the discussion

The discussion begins with the statement that „WC-adjusted elevated BMI was attributed to an increased bone density, limb muscle mass, or subcutaneous fat“.. But in your study, you did not make any measurements to support this claim. No dynamometer ,bone densitometry, no CT, or Short Physical Performance Battery test

Line 375 you describe the mRNA synthesis of acyl-CoA, a key ENZYME for adipogenesis!!!!!!!!!

and inflammation, but in your work you described none of the pro-inflammatory parameters.

and so I could describe paragraph after paragraph.

A lot of the discussion is about adiponectin. Why, when you did not measure adiponectin at all?

Please rewrite the discussion and specifically focus on comparing only the results that are presented in your work

Author Response

Response to Reviewer 2 Comments

Point 1: I'm a bit confused about the results...table 1.1 shows BMI levels...............but those aren't BMI levels, are they? Subsequently, the first line states number....numbers of what? A number of participants in the correct group? 56,69????,....sex, male?23,756? What is it?

Please explain the individual items in the table more clearly... similarly, table 1-2 and others.

line 289, why is there under the table abbreviation: WC?

Response 1: Firstly, thank you for your time to review our manuscript.

We apologize for the confusion. BMI levels 1, 2, 3, 4, and 5 written horizontally at the top of Table 1-1 are the five BMI levels, as explained in the methods section. The first line on the left side of the table shows the number of the participants and the percentage is shown in parentheses. For example, among all participants, BMI level 1 corresponds to 56,693 (7.5%) participants, and BMI level 2 corresponds to 313,840 (41.4%) participants. The second line on the left side of the table shows the number of male participants, who are 39.25% in proportion in all participants. BMI level 1 is exhibited in 23,756 (41.9%) males. We have reflected your comment and for being clearer, we have changed Sex, Male (39.25) to Sex, Male (39.25%). Thank you for your insights. I hope we have understood your intention regarding table 1-2 and other tables. We have provided explanations for abbreviations below every table. If we have misunderstood your intention, please let us know.

Point 2: The weakest part of the article is the discussion

The discussion begins with the statement that „WC-adjusted elevated BMI was attributed to an increased bone density, limb muscle mass, or subcutaneous fat“.. But in your study, you did not make any measurements to support this claim. No dynamometer ,bone densitometry, no CT, or Short Physical Performance Battery test

Line 375 you describe the mRNA synthesis of acyl-CoA, a key ENZYME for adipogenesis!!!!!!!!!

and inflammation, but in your work you described none of the pro-inflammatory parameters.

and so I could describe paragraph after paragraph.

A lot of the discussion is about adiponectin. Why, when you did not measure adiponectin at all?

Please rewrite the discussion and specifically focus on comparing only the results that are presented in your work.

Response 2: You have raised an important point. Water retention can occur beside bone, muscle, or subcutaneous fat. However, as shown in tables 1-1 and 1-2, ESRD occurred in only 0.106% (in 805 out of 58,471 participants) of the participants. We have reflected on your comment by adding information on water retention in the dicussion. After adjusting for WC, i.e., after excluding abdominal obesity, the factors that have a large variation range in affecting BMI are bone density, limb muscle mass, and subcutaneous fat. We believe that this is not a claim but a big premise.

Please consider our following explanation. As presented in section 2. Materials and Methods (line 68), this is a big data analysis research based on a national database created by health examination and conducted by the Korean government for the entire nation. Therefore, as the participants are anonymous, we cannot change or revise the health examination records. This paper aims to derive a meaningful result from analyzing the national database and claims in the discussion that our result is scientifically reasonable by quoting several previous classic studies. Previously published big data analysis research cannot be meaningless. As an example quoted in your review, we mentioned adiponectin in this paper not to identify the relationship among adiponectin, obesity, and MI but to claim that our result (HR for MI has a positive relationship with WC and not with BMI in old age) is scientifically reasonable by quoting already published classic studies.

Lastly, we appreciate your insights to impove our manuscript.

Reviewer 3 Report

This paper focused on building prediction model of myocardial infarction (MI) using body mass index (BMI) and waist circumference (WC). This study use a good data which is Korea National Health Insurance System (NHIS). There are several queries need to be addressed before reviewer can accept this paper.

Revision

1.     In Line 75-76, authors mentioned that they include participants from 2009-2012. Is there any specific reason why authors only included patients from that period? Because it is possible for a change in lifestyle, economy, etc., during 10 years from 2012-2022.

2.     This is a retrospective or prospective cohort study? Please add this information on method.

3.     In line 124, authors mentioned that each participant had anthropometry during their physical examination. Is it done during their first day in hospital or based on medical record?

4.     In line 127-128, authors mentioned variables such as heavy drink and physical activity. Is this two variables also available in medical record?

5.     In line 150, please add more detailed summary of table 1-2.

6.     Please reduce your table size, only include 4-5 important tables in manuscript and put the rest on appendix or supplement.

7.     Because authors aims to build prediction model, please emphasize the most significant prediction model in this study.

Author Response

Response to Reviewer 3 Comments

Point 1: In Line 75-76, authors mentioned that they include participants from 2009-2012. Is there any specific reason why authors only included patients from that period? Because it is possible for a change in lifestyle, economy, etc., during 10 years from 2012-2022.

Response 1: Firstly, thank you for your time to review our manuscript.

As mentioned on lines 80-81, we conducted F/U for an average of 6.98 years until December 2018 for individuals who had undergone a health examination from 2009 to 2012. That is, we finished F/U in December 2018, and the data were obtained after that for approximately 1 year. Regarding the reason for starting in 2009, the NHIS started including WC measurements since 2009. As you pointed out in point 2, adding ‘prospective cohort study’ in the methods section would make the explanation clearer. Thank you for your insights. 

Point 2: This is a retrospective or prospective cohort study? Please add this information on method.

Response 2: As mentioned in response for point 1, we agree on adding that information in the methods section. We have reflected your comment by adding ‘prospective cohort study’ in the methods section.

Point 3: In line 124, authors mentioned that each participant had anthropometry during their physical examination. Is it done during their first day in hospital or based on medical record?

Response 3: We believe you are aware of the Korean National Health Care System. It is famous for its efficiency and broad coverage for almost the entire Korean papulation (even immigrants), although problems do exist. Suprisingly, all health examinations for every adult over 20 years of age every 2 years are conducted in just one day and no hospitalization is required. The health examination includes a questionnaire, anthropometric measurements, blood pressure measurement, eyesight analysis, hearing analysis, chest radiograph, basic hematological investigations (blood glucose level, AST/ALT, γ-GTP, lipid profile, Hb level, creatinine concentration, etc), basic urine examination using a stick, cervical pap smear test (for woman), and gastroduodenoscopy (for individuals aged over 40 years). It requires a minimum of 30 minutes and a maximum of 3 to 4 hours. We have reflected your comment by adding this sentence in section 2.1 Data collection. Thank you for your advice. Further, we have mentioned on lines 131-132 that anthropometric measurements were obtained during physical examination by trained specialists on the day of health examination.

Point 4 : In line 127-128, authors mentioned variables such as heavy drink and physical activity. Is this two variables also available in medical record?

Response 4: The health examination includes a self-adminstered tionnaire including questions pertaining to health-related life behavior, which is filled by the examinee. Questions pertaining to heavy drinking include “How many times a week do you drink? and When you drink, what kind of alcohol do you drink and how many drinks do you have?” and questions pertaining to physical activity include “In the past week, on how many days did you engage in a strenuous activity that made you out of breath for more than 20 minutes a day (e.g., running, aerobics, brisk biking, mountain climbing, etc.)? In the past week, on how many days did you engage in moderate activity that made you breathe a little harder than usual for 30 minutes or more a day (e.g., walking briskly, playing doubles’ tennis, biking at a normal speed, mopping face down, etc.)? and In the past week, on how many days in total did you walk for at least 30 minutes per day, including at least 10 minutes at a time (e.g., light exercise, including walking to work or leisure time)?” Therefore, we obtained information pertaining to these two variables from the questionnaire filled by the examinee. We have added a sentence in section 2.6 Definition of variables to reflect your comment. Thank you for your insight.

Point 5 : In line 150, please add more detailed summary of table 1-2.

Response 5: We had presented a summary of the result in section 2.7 Measured outcomes. However, we agree with you. We have deleted that part and moved it to the results section. Thank you for your suggestion.

Point 6 : Please reduce your table size, only include 4-5 important tables in manuscript and put the rest on appendix or supplement.

Response 6: We have reflected your comment by including tables 5-1, 5-2, 6-1, and 6-2 as supplementary files. Another reviewer also pointed out that the tables duplicate the figures. Thank you for your kind advice.

Point 7 : Because authors aims to build prediction model, please emphasize the most significant prediction model in this study.

Response 7: We agree with you and thank you for your insight. We have reflected your advice by adding some lines in the conclusion section.

Lastly, we appreciate your time and effort to review our paper. Thank you.